# Potential Applications of an Exopolysaccharide Produced by *Bacillus xiamenensis* RT6 Isolated from an Acidic Environment

**DOI:** 10.3390/polym14183918

**Published:** 2022-09-19

**Authors:** Elisa Huang-Lin, Enrique Sánchez-León, Ricardo Amils, Concepcion Abrusci

**Affiliations:** 1Departamento de Biología Molecular, Facultad de Ciencias, Universidad Autónoma de Madrid (UAM), Cantoblanco, 28049 Madrid, Spain; 2Centro de Biología Molecular Severo Ochoa, Consejo Superior de Investigaciones Científicas, Universidad Autónoma de Madrid (UAM), 28049 Madrid, Spain

**Keywords:** exopolysaccharide, *Bacillus*, antioxidant, chelating, emulsifying

## Abstract

The *Bacillus xiamenensis* RT6 strain was isolated and identified by morphological, biochemical and molecular tests from an extreme acidic environment, Rio Tinto (Huelva). Optimisation tests for exopolysaccharide (EPS) production in different culture media determined that the best medium was a minimal medium with glucose as the only carbon source. The exopolymer (EPS_t_) produced by the strain was isolated and characterised using different techniques (GC-MS, HPLC/MSMS, ATR-FTIR, TGA, DSC). The molecular weight of EPS_t_ was estimated. The results showed that the average molecular weight of EPS_t_ was approximately 2.71 × 10^4^ Da and was made up of a heteropolysaccharide composed of glucose (60%), mannose (20%) and galactose (20%). The EPS_t_ showed antioxidant capabilities that significantly improved cell viability. Metal chelation determined that EPS_t_ could reduce the concentration of transition metals such as iron at the highest concentrations tested. Finally, the emulsification study showed that EPS_t_ was able to emulsify different natural polysaccharide oils, reaching up to an 80% efficiency (olive and sesame oil), and was a good candidate for the substitution of the most polluting emulsifiers. The EPS_t_ was found to be suitable for pharmaceutical and industrial applications.

## 1. Introduction

Over the last few years, the screening of microorganisms and their microbial exopolysaccharides (EPSs) has received increasing attention as a high-value product [1,2]. Unlike plant and animal polymeric products, the EPSs of different microbial origins present a large diversity in structural composition (xanthan, dextran, alginate, cellulose, hyaluronic acid), which provide them with distinctive properties for industrial applications [3]. 

Research on extremophilic microorganisms has gained particular attention [4] as a production source for new biomolecules [5]. Since these microorganisms are known to develop in extreme environments, their biopolymers are expected to have unique properties to adapt to such conditions [6]. A potential use case can be found in pharmaceutical industries to replace chemical antioxidants that can cause long-term damage and have safety concerns [7]. Some polysaccharides are reported to enhance the defense mechanism of cells, aside from reducing the oxidative damage caused by reactive species (ROS) or preventing free radical accumulation and inhibiting lipid peroxidation in vitro [8,9]. Moreover, toxic levels of heavy metals may interact with important biomolecules in the cell, including DNA and protein, leading to the excessive production of ROS [10]. EPSs are applied as radical scavengers to protect the body from free radicals that can cause different chronic diseases [11]. Currently, there are studies reporting the antioxidant activity of *Bacillus* EPS, such as *Bacillus licheniformis* [12], *Bacillus subtilis* [13] or *Bacillus amyloliquefaciens* [14]. EPSs are also reported as a source of metal chelating compounds [15], as it is known that metal ions can generate highly reactive oxygen free radicals by Fenton or Haber–Weiss reactions. These reactive hydroxyl radicals can interact with many biological molecules, leading to lipid peroxidation, DNA damage and protein denaturalisation [16]. Previous studies showed *Bacillus* strains with metal chelating properties, such as *Bacillus haynesii* [15], *Bacillus firmus* [17] and *Bacillus cereus* [18]. In addition, numerous studies have been performed to search for biomedical applications [19,20,21,22]. In industrial applications, emulsifiers based on polysaccharides of microbial origin have attracted considerable attention in recent years [23] as an alternative to chemical-based emulsifiers, which are extensively used in various industries and have raised concerns associated with their toxic effects on the environment [24], e.g., *Bacillus mojavensis* [25] or *Bacillus albus* [26].

On the other hand, Rio Tinto is considered as an extreme environment exhibiting a constant and very acidic pH (2.5) buffered by ferric iron and with high concentrations of heavy metals. These extreme conditions originate from the metabolic activity of chemolithotrophic prokaryotes that are found in high numbers in its waters [27].

This study aims to describe the isolation and identification using the morphological, biochemical and molecular biology of the *Bacillus xiamenensis* RT6 strain isolated from an extremely acidic environment, Río Tinto (Huelva, Spain). In addition to EPS production optimisation, molecular weight estimation and characterisation by GC-MS, HPLC/MSMS, ATR-FTIR, TGA and DSC, its biotechnological applications were evaluated. This species has been described in agricultural applications as a potential plant-growth-promoting rhizobacteria (PGPR) antagonist [28] and biocontrol agent [29]. However, the strain is novel and its biotechnological applications have not been studied. The hypothesis of this work is that *Bacillus xiamenensis* RT6 may have a wide range of potential biotechnological applications, such as pharmaceutical and industrial use. Therefore, the present study was carried out to test different capacities, including anti-oxidation, metal chelation, cell viability and emulsification properties.

## 2. Materials and Methods

### 2.1. Chemicals and Standards

Antibiotic discs were purchased from BBL^TM^ Sensi-Disc^TM^ (Sensi-Disc^TM^, Fisher Scientific, Waltham, MA, USA). Microbial strains were obtained from the Colección Española de Cultivos Tipo (CECT, Madrid, Spain). The HeLa cells line was purchased from CLS (Cell Line Service, Eppelheim, Germany). The API gallery was obtained from the rest of experimental reagents, which were purchased from BioMérieux, Madrid, Spain, or of analytical grade. 

### 2.2. Bacterial Strain and PCR Amplification

The indigenous bacterial strain RT6 was isolated from the sediments of the river source in Río Tinto (Huelva), Spain. 37°43′19″ N 6°33′03″ W. Triplicate samples (1 g) of sediment were homogenised in 10 mL of NaCl 0.6 M, and serially diluted (10 folds). Aliquots of 100 μL were spread on trypticase soya agar (TSA) plates (Sigma–Aldrich Ireland Limited, Arklow, Ireland) and incubated overnight at 30 °C. The strain isolate was transferred to new TSA plates and stored at −80 °C in 30% glycerol. Morphological and biochemical tests were carried out for the strain characterisation. Bacterial motility and sporulation tests were performed by using a microscope (B100iMSa, Madrid, Spain). Gram stain was determined following the operative procedure described by Beveridge [30]. The biochemical characterisation was carried out using an API gallery (BioMérieux, Madrid, Spain). The antibiogram test was carried out following the method described by Blanc et al. [31] in order to determine the sensitivity of the bacteria with respect to antibiotics from different commercial discs (BBL^TM^ Sensi-Disc^TM^, Fisher Scientific, MA, USA): nalidixic acid 30 μg; cephalothin 30 μg; chloramphenicol 30 μg; erythromycin 15 μg; streptomycin 10 μg; tetracycline 30 μg; trimethoprim 5 μg. A TSA plate was inoculated with an inoculum of 2.5 × 10^7^ CFU/mL and incubated at 30 °C for 24 h. The degree of sensitivity to each antibiotic was classified based on the following criteria: sensitive (S): halo diameter greater than 15 mm; resistant (R): halo diameter less than 13 mm; intermediate (R/S): halo diameter between 13 and 15 mm.

PCR amplification was undertaken as described by Morro et al. [32]. DNA from the isolated microorganism was extracted using a FastDNA kit for soil (BIO 101, Vista, CA, USA) and purified with JetQuick^®^ Genomic DNA Purification Kit (Genomed, Leesburg, USA) following manufactural indications. Extracted DNA was used as a template to amplify the 16S rRNA gene region of the isolated microorganism. PCR reactions were carried out in 50 μL of final volume using the primer pairs: 27F (5′–AGA GTT TGA TC(C/A) TGG CTC AG−3′) and 1492R (5′–TAC GG(CT) TAC CTT GTTACG ACT T–3′). Amplifications were carried out in a Cycler 2720 Thermocycler (Applied Biosystems, Foster City, CA, USA) set as follows: 5 min at 94 °C; 30 cycles including a phase at 94 °C for 1 min, another step at melting temperature for 30 s and the last one at 72 °C for 3 min. Finally, an extension step was performed at 72 °C for 10 min. PCR products were separated by agarose gel electrophoresis and visualised under UV-light, after staining, with Greensafe (NZYtech, Lisboa, Portugal). 

PCR products were sequenced and compared with data from GenBank database using BLAST (National Centre of Biotechnology, Bethesda, MD, USA) to identify the closest sequences. Selected sequences were aligned with CLUSTAL_X software program (version 1.8)(Conway Institute UCD, Dublin, Ireland) [33]. 

### 2.3. Analysis of Bacterial Growth, Colony Forming Units (CFU)/mL, pH and EPS Production 

To optimise cell growth, standard culture conditions were compared with two different mediums, a minimum growth medium (MGM) and an enriched medium. These conditions were established by transferring the strain, previously inoculated in TSA medium and incubated at 30 °C for 24 h, to a 250 mL flask filled with 50 mL of the different growth mediums. Minimal growth medium (MGM) was prepared as described by Abrusci et al. [34]: g/L: K_2_HPO_4_ 0.5, KH_2_PO_4_ 0.04, NaCl 0.1, CaCl_2_·2H_2_O 0.002, (NH_4_)_2_SO_4_ 0.2, MgSO_4_·7H_2_O 0.02, FeSO_4_ 0.001, and glucose as a carbon source at a concentration of 4 g/L. The enriched medium contained in g/L: casein peptone 17.0, soya peptone 3.0, NaCl 5.0 and glucose 2.5. 

The inoculum was prepared at an optical density of OD_550 nm_ (2.5 × 10^7^ CFU/mL) with a spectophotometer Biowave II + (Biochrom Cambridge, UK). In order to determine when the culture reached the beginning of its stationary phase, samples were measured at 24, 48, 72 and 96 h of incubation. The cell growth number was evaluated by different plating dilutions incubated at 30 °C for 24 h with TSA agar medium. A Thermo Orion pH Meter Model 2Star (Thermo Scientific, Asheville, NC, USA) was used to determine the pH values during each measurement period. 

For determination of EPS production, the strain was inoculated from the stock culture in TSA medium and incubated at 30 °C for 24 h. After that, the strains were transferred into flasks of 250 mL filled with 50 mL of different mediums (MGM or enriched medium). The flasks were incubated in a rotary shaker incubator (model Orbitek LJEIL; Scigenics Biotech, Bangalore, India) at 30 °C and 130 rpm for 24 h. After the first incubation, 10 mL of this broth (2.5 × 10^7^ CFU/mL concentration) was inoculated into flasks of 2000 mL filled with 1000 mL of MGM supplemented with glucose to improve EPS productivity. The flasks were incubated at 30 °C and 130 rpm for 24 h, when the stationary phase was reached. Three independent assays were performed. The EPSs were isolated and quantified for each treatment.

### 2.4. Isolation, Purification and Molecular Weight of EPS

Isolation and purification of EPS was performed following the method of Sánchez-León et al. [21]. The cultures obtained from the strain were centrifuged at 13,154× *g* for 30 min at 4 °C with a DuPont Sorvall RC−5 centrifuge (Wilmington, DE, USA). The EPS was precipitated with cold ethanol 96% (three times the volume) and left overnight. The precipitate was collected by centrifugation at 13,154× *g* for 30 min at 4 °C and dissolved in Milli-Q water. Then, the crude EPS was dialysed at 4 °C with Milli-Q water for 48 h. The dialysed contents were freeze–dried by lyophilisation with a Flexy–Dry MPTM freeze dryer (FtS Systems Inc., Stone Ridge, NY, USA) for 48 h, and the dry weight of the powdered EPS was determined.

For the purification of the EPS, the obtained EPS (10 mL, 10 mg/mL) was subjected to a DEAE−52 anion exchange column (Aldrich Chemical Company, Inc., Milwaukee, WI, USA) (2.6 × 30 cm) and eluted with deionised water. Concentrations of 0.2, 0.5, 1.0 and 1.5 M of NaCl were used as eluent at 1 mL/min flow rate. The eluents (10 mL/tube) were monitored by phenol–sulphuric acid method [35]. The fractions were collected, concentrated and lyophilised, and the obtained EPS was named EPS_t_.

### 2.5. Compositional Analysis and Characterisation of the EPS

#### 2.5.1. Compositional Analysis 

Molecular weight of the purified fraction was determined by gel filtration chromatography [22]. Standard reference dextrans of 5, 12, 50 and 80 KDa (Sigma, St. Louis, MO, USA) molecular weight were used. A Sephadex G−100 column (Aldrich Chemical Company, Inc., Milwaukee, USA) (1.6 × 50 cm) eluting with 0.2 mol^−1^ NaCl solution at flow rate of 1 mL/min was used for this experiment. The molecular weight of EPS_t_ was derived from the standard plot of the reference dextrans.

To determine the monosaccharide composition prepared as described in the literature [21], EPS_t_ was hydrolysed with 0.5 M trifluoroacetic acid (TFA) Aldrich^®^ (Schnelldorf, Germany) at 120 °C for 2 h. Samples were treated before and after the process with N_2_. The derivative products were used for the determination of the monosaccharide composition by gas chromatography coupled with mass spectrometry detector (GC–MS). An EVOQ GC–TQ Bruker (Bruker, Billerica, MA, USA) gas chromatography system was employed. A total of 1 µL of samples was injected in the ratio of 100:1 in splitless mode with source temperature at 230 °C. The separation was held in Rxi^®^−5Sil MS (Restek Corporation, Bellefonte, PE, USA), capillary column having 30 m length × 0.250 mm the width and 0.25 µm, with helium as carrier gas at a constant flow rate of 1 mL/min. The initial temperature was 50 °C at a hold time of 2 min and was followed by increasing to 280 °C at 10 °C ramp rate with 5 min hold time. The trimethylsilylated mono–sugars such as glucose, arabinose, xylose, mannose, galactose, fructose, galacturonic acid and glucuronic acid were used as standards. The composition of amino acids and uronic acids was determined by HPLC/MSMS (Agilent Technologies 1100 series–6410B (TQ), Waldbronn, Germany). An ACE Excel 3 C18–Amide column (Advanced Chromatography Technology LTD, Aberdeen, Scotland) as a stationary phase was used with a mobile phase of 0.1% formic acid in water. Flow rate was 0.2 mL/min. The temperature for analysis was set at 40 °C.

#### 2.5.2. Attenuated Total Reflectance/FT-Infrared Spectroscopy (ATR-FTIR)

The EPS_t_ structural–functional groups were detected using attenuated total reflectance/FT–infrared spectroscopy (ATR–FTIR). IR spectra were obtained using a BX–FTIR spectrometer (Perkin Elmer, Waltham, MA, USA) coupled with an ATR accessory, MIRacle^TM^ –ATR (Pike Technologies, Cottonwood, AZ, USA) and spectra were obtained from 32 scans at 4 cm^–1^ of resolution from 400 to 4000 cm^–1^.

#### 2.5.3. Thermogravimetric (TGA) and Differential Scanning Calorimetry (DSC) Analysis

Thermogravimetric analysis (TGA) was conducted in a TGA Q500 (TA Instruments, New Castle, DE, UK) equipped with an EGA oven and operating at atmospheric pressure. EPS_t_ sample (1–3 mg) was placed in a Pt crucible and heated at a rate of 10 °C/min over a temperature range from 20 °C to 800 °C in air (90 mL/min) atmosphere. Data were processed using TA Universal Analysis software (TRIOS 5.2). Differential scanning calorimetry (DSC) was performed on DSC Q100 (TA Instruments, New Castle, DE). The calorimeter was previously calibrated and certified by the National Institute of Standards and Technology NIST. A total of 0.5–2 mg of dried EPS_t_ sample was placed in an aluminium pan without lid. Then, it was analysed using an empty pan as a reference and 50 mL/min air purge gas. The heating rate was 10 °C/min from 20 °C to 600 °C.

### 2.6. Antioxidant Activity Tests 

The free radical scavenging activities for 1,1–diphenyl–2–picryl–hydrazyl radical (DPPH•), hydroxyl radical (•OH) and superoxide anion (O_2_^−•^) were assessed as indicators of the antioxidant activity of the EPS_t_. Absorbances were measured using a FLUOstar Omega BMG LABTECH (Aylesbury, UK) spectrophotometer. 

#### 2.6.1. DPPH (1,1–Diphenyl−2–picryl–hydrazyl) Free Radical Scavenging Activity 

The scavenging activity for DPPH was assayed as described in the study by Niknezhad et al. [36]. The reaction mixture contained 50 μL of EPS_t_ at different concentrations (0.1, 0.25, 1.0, 2.5, 5.0, 7.5 and 10 mg/mL) and 100 μL of DPPH (100 μM DPPH–ethanolic solution) (Sigma Chemical, St Louis, MO, USA). The mixtures were shaken vigorously and incubated in the dark at 25 °C. After 30 min, the absorbance was recorded at OD_525nm_. Ascorbic acid (Vc) (Sigma Chemical, St Louis, MO, USA) was used as positive control.

The percentage of radical scavenging activity for DPPH was calculated according to Formula (1): DPPH scavenging activity [%] = [ 1 − (A_1_ − A_2_)/A_0_] × 100](1)
where

A_1_ = OD_525 nm_ of reaction mixture.

A_2_ = OD_525 nm_ of reaction mixture without DPPH. 

A_0_ = OD_525 nm_ of the reaction mixture with DPPH but without EPS_t_. 

#### 2.6.2. Hydroxyl Radical Scavenging Activity 

The scavenging activity for hydroxyl radicals was assayed using the FeSO_4_–salicylic acid method as described by Sun et al. [37]. The reaction mixture contained 40 µL of FeSO_4_ solution (9 mM), 40 µL of salicylic acid (9 mM ethanol–salicylic acid solution) (Sigma Chemical, St Louis, MO), 40 µL of EPS_t_ at different concentrations (0.1, 0.25, 1.0, 2.5, 5.0, 7.5, and 10 mg/mL) and 40 µL of H_2_O_2_ (8.8 mM) (Sigma Chemical, St Louis, MO, USA). Then, the mixtures were incubated at 37 °C. After 30 min, the absorbance at OD_510nm_ was measured. Ascorbic acid (Vc) was used as positive control. 

The percentage of hydroxyl radical scavenging activity was calculated according to Formula (2): Hydroxyl radical scavenging activity [%] = [ 1 − (A_1_ − A_2_)/A_0_] × 100](2)
where

A_1_ = OD_510nm_ of the reaction mixture.

A_2_ = OD_510nm_ of the reaction mixture without salicylic acid. 

A_0_ = OD_510nm_ of the reaction mixture with salicylic acid but without EPS_t_.

#### 2.6.3. Superoxide Anion Scavenging Activity 

The superoxide scavenging activity was determined according to the method described by Balakrishnan et al. [38]. The reaction mixture contained 0.3 mL of EPS_t_ at different concentrations (0.1, 0.25, 1.0, 2.5, 5.0, 7.5 and 10 mg/mL), 2.6 mL of phosphate buffer (50 mM, pH 8.2) and 90 μL of pyrogallol (3 mM) (Sigma Chemical, St Louis, MO, USA) dissolved in HCl (10 mM). Then, the absorbance was measured from 0 min to 10th min at OD_325nm_. Ascorbic acid (Vc) was used as positive control. 

The percentage of superoxide scavenging activity was calculated by Formula (3): Superoxide scavenging activity [%] = 1 − [(A_10_/C_10_) − (A_0_/C_0_)] × 100(3)
where 

A_0_ and A_10_ = OD_325nm_ of the reaction mixture at 0 min and 10 min.

C_0_ and C_10_ = OD_325nm_ of the reaction mixture without pyrogallol at 0 min and 10 min.

#### 2.6.4. Metal Ion Chelating Activity 

The chelating ability of metal ion was measured according to the method reported by Li et al. [7] in terms of chelating ferrous ion (Fe^2+^) in the iron–ferrozine complex. The activity was measured at two different final reaction mixture pH values (2.5 and 5.6). Briefly, the reaction mixture, containing 1.0 mL of EPS_t_ sample solution (0.1, 0.25, 1.0, 2.5, 5.0, 7.5 and 10 mg/mL), 0.05 mL of FeCl_2_ solution (2 mM), 0.2 mL of ferrozine solution (5 mM) (Sigma Chemical, St Louis, MO, USA) and 2.75 mL of Milli–Q water, was shaken well and incubated at room temperature for 10 min. Final pHs (2.5 and 5.6) of the reaction mixtures were adjusted with HCl or NaOH. The absorbance of the mixture was measured at OD_562nm_. Ethylenediaminetetraacetic acid (EDTA) (Sigma Chemical, St Louis, MO, USA) was used as the positive control. 

The chelating ability on ferrous ion was calculated according to Formula (4):Chelating ability [%] = [(A_0_ − (A_1_ − A_2_)/A_0_)] × 100(4)
where,

A_0_ = OD_562nm_ of deionised water. 

A_1_ = OD_562nm_ of the reaction mixture. 

A_2_ = OD_562nm_ of the reaction mixture but without FeCl_2._

### 2.7. Biocompatibility Studies

#### 2.7.1. Culture of Cells

HeLa cells (human T–cell lymphoblast-like cell line) were chosen as a reference cell line to study EPS_t_ biocompatibility. HeLa cells were cultured in Dulbecco’s modified Eagle’s medium (DMEM) (Sigma Chemical, St Louis, MO, USA) supplemented with 10% fetal bovine serum (FBS) (Sigma Chemical, St Louis, MO, USA), 2 mM L–glutamine (Sigma Chemical, St Louis, MO, USA), penicillin (100 IU/mL) (Sigma Chemical, St Louis, MO, USA) and streptomycin (100 μg/mL) (Sigma Chemical, St Louis, MO, USA) under 5% CO_2_ atmosphere and at 37 °C for different times. 

#### 2.7.2. Cytotoxicity Assay

Cytotoxicity of the exopolysaccharide was assessed by the reduction in the MTT reagent (3–[4,5–dimethyl-thiazol−2–yl]−2,5–diphenyltetrazoliumbromide) to formazan (GE Healthcare, Uppsala, Sweden) following the method proposed by Tada et al. [39]. HeLa cells were seeded in a 24–well culture plate (5 × 10^5^ CFU/mL) and 100 µL of different concentration of EPS_t_ (0, 25, 50, 100, 200, 400 µg/mL) was transferred into each well for 24 h treatment. The absorbance at OD_590nm_ was measured using a microplate reader (LT−4000, Labtech International Ltd, Lewes, UK).

### 2.8. Determination of the Antioxidant Ability on Cellular Level

#### 2.8.1. Establishment of Injury Model 

The establishment of injury model against HeLa cells was assayed following the methods described by Zhou et al. [40]. HeLa cells density was readjusted to 5 × 10^4^ CFU/well and the suspension was seeded into a 96–well plate for 24 h. After this period, solutions were removed and 100 µL of different concentrations of H_2_O_2_ (0.25, 0.5, 1, 2 mM) was added into the plate for 1 h under 5% CO_2_ atmosphere incubator and at 37 °C. After exposure, all solution was removed, fresh medium was added into the plate, and the cell viability was determined with the MTT method (GE Healthcare, Uppsala, Sweden). HeLa cells viability was calculated according to Formula (5): Cell viability [%] = (A_1_/A_2_) × 100(5)
where 

A_1_ = absorbance at OD_590nm_ of the cells with MTT solution treated previously with H_2_O_2_.

A_2_ = absorbance at OD_590nm_ of the cells with MTT solution without any treatment. 

#### 2.8.2. Determination of the Protection of EPS_t_ on HeLa Cells against Oxidative Stress

The determination of the protection capacity of EPS_t_ on HeLa cells against oxidative stress was assayed following the methods described by Zhou et al. [40]. HeLa cells were seeded into 96–well plate for 24 h at a concentration of 5 × 10^4^ CFU/well. After this time, DMEM solutions were removed and EPS_t_ diluted in DMEM at different concentrations (25, 50, 100, 200, 400 µg/mL) were added into each well for 1 h treatment. The solutions with EPS_t_ were removed and a new medium with a high concentration of H_2_O_2_ (2 mM) was added and incubated for 1 h. Ascorbic acid (20 mg/mL) was used as a positive control. Finally, cell viability was determined with MTT method (GE Healthcare, Uppsala, Sweden). HeLa cells viability was calculated according to Formula (6): Cell viability [%] = (A_1_/A_2_) × 100(6)
where 

A_1_ = absorbance at OD_590nm_ of the cells with MTT solution treated previously with H_2_O_2_ and EPS_t_.

A_2_ = absorbance at OD_590nm_ of the cells with MTT solution without any treatment. 

### 2.9. Emulsifying Properties 

The emulsifying properties of EPS_t_ were evaluated using the method by Meneghine et al. [41]. The assays were undertaken in transparent cylindrical 5 mL tubes that contained 1.5 mL of an oil phase and 1.5 mL of an aqueous phase. The oil phase contained vegetable oils (sunflower oil, olive oil, sesame oil and coconut oil). For the aqueous phase, both commercial emulsifying compounds, such as polysorbate 20 (Tween 20) (Sigma Chemical, St Louis, MO, USA), sodium dodecyl sulphate (SDS) (Sigma Chemical, St Louis, MO, USA), Triton X−100 (Sigma Chemical, St Louis, MO, USA) and the obtained polymers were used for comparison purposes. All compounds used had a concentration of 3:2 *v/v*. The tubes were stirred in a vortex at 2400 rpm for 2 min. After 24 h and 168 h, the emulsification indexes E_24_ and E_168_ were determined with Formula (7): E [%] = HEL/HT × 100(7)
where HEL (mm) is the height of the emulsion layer and HT (mm) is the overall height.

### 2.10. Statistical Analysis

All experiments were performed in triplicate. Analysis of variance test (ANOVA) was performed to make the statistical comparisons by using the Statistical Package for the Social Sciences version 21 (SPSS^®^ Inc., Chicago, IL, USA). *p* < 0.05 was considered statistically significant.

## 3. Results and Discussion 

### 3.1. Strain Identification and EPS Optimisation

Morphological studies of the isolate indicated that it was a sporulated Gram–positive *Bacillus* with movement capacity. Biochemical tests showed the presence of beta–galactosidase, tryptophan deaminase, gelatinase and catalase enzymatic activities. A citrate utilisation capacity was also detected. The isolate showed sensitivity to chloramphenicol, cephalothin, tetracycline, nalidixic acid, enthromycin, streptomycin and trimethoprim. Moreover, a positive catalase activity was detected. 

The bacterial strain was identified by means of the 16S rRNA sequence obtained after PCR amplification and sequencing. A comparison of the 16S rRNA sequences of the isolated strain with the sequences available in the GenBank database showed that the isolated bacteria had a 97.67% similarity to *Bacillus xiamenensis* (accession number MK358984.1). *B. xiamenensis* was previously reported for its heavy metal tolerance and role in the assisted phytoextraction of Cr-contaminated soils in association with the *Sesbania* plant [42]. Thereby, the presence of this bacterium could be expected in the isolated site since it is known that Rio Tinto is one of the most acid-rock drainage fluvial–estuarine systems in the world, and can reach a total Fe concentration of up to 20 g/L [43,44]. 

Cellular growth, pH values, EPS optimisation and purification are represented in Figure 1. The growth of *B.xiamenensis* RT6 with glucose as the only carbon source, pH values and exopolymer production at 30 °C are shown in Figure 1a. Cell growth reached its maximum (7.45 log CFU/mL) after 24 h. The maximum production of EPS, 748 mg/L, occurred at 24 h and pH 7. During the process, no acute drop in pH was detected. On the other hand, the growth of *B. xiamenensis* RT6 in an enriched medium is shown in Figure 1b, where cell growth reached its maximum (11.30 log CFU/mL), with a maximum EPS production of 12.2 mg/L after 48 h at pH 7.6. The results showed that optimal bacterial growth was obtained with the enriched medium. However, the highest production of EPS was obtained with the minimum medium: glucose as the only carbon source. This maximum production of EPS with glucose took place at the beginning of the stationary phase. Similar results were obtained with *Pseudomonas* strains [45], also confirming that pH 7 was optimal for EPS production by *Bacillus pseudomycoides* [46]. On the other hand, the production of EPS (748 mg/L) by *B. xiamenensis* RT6, with glucose as the only carbon source, was even higher than those obtained by *B. licheniformis* AG−06, with an EPS production of 556 ± 0.18 mg/L [47]. EPS production by *B. xiamenensis* RT6 was even more remarkable compared to *Bacillus circulans*, (65 mg/mL) using sucrose and yeast as energy sources at longer incubation times (96 h) [48]. This indicated that an enriched media decreased EPS production.

In addition, glucose, as the only carbon source introduced in this medium, is considered to be of low cost and easy to acquire for biotechnological applications due to its high productivity and yield [49,50]. 

EPS obtained from glucose as an energy source was purified (EPS_t_) and showed a single peak, characteristic of high-purity exopolysaccharides (Figure 1c). 

### 3.2. Compositional Analysis and Characterisation of EPS_t_ Exopolymer 

#### 3.2.1. Molecular Weight Determination for EPS_t_

Molecular weight estimation for EPS_t_ is represented in Figure 2. The estimated molecular weight of the purified EPS_t_ was approximately 2.71 × 10^4^ Da (Figure 2a), which was calculated from the dextran standards calibration curve formula (Figure 2b). It is reported that the average molar mass for heteropolysaccharides ranges between 4 × 10^4^ and 6 × 10^6^ Da. In this case, EPS_t_ presented a lower molar mass in comparison with most of the reported heteropolysaccharides [51]. Studies showed that the average molecular weight of an EPS produced by freshwater dynamic sediment-attached *Bacillus megaterium* RB−05 [52], composed of glucose, galactose, mannose, arabinose, fucose and N–acetyl glucosamine, was found to be 1.7 × 10^5^ Da. Moreover, EPS isolated from *Bacillus coagulans* RK−02 [53], composed of galactose, mannose, fucose and glucose, had an average molecular weight of 3 × 10^4^ Da.

#### 3.2.2. Gas Chromatography GC-MS Analysis for EPS_t_

Gas chromatography GC–MS analysis (Figure 3) showed four peaks, which were identified as the monosaccharides glucose (α–glucose, β–glucose), β–mannose and α–galactose, representing 60%, 20% and 20% of the total monosaccharide content, respectively. This indicates that EPS_t_ is a heteropolysaccharide. In addition, the results obtained in the determination of amino acids and uronic acids by HPLC confirmed the absence of both found in the sample. Other *Bacillus* strains, such as the EPS produced by *Bacillus velezensis* SN−1 with antioxidant and antitumor properties, reported the presence of mannose and glucose representing approximately 54.60% and 45.30%, respectively, with an absence of galactose [54]. This variability in content can be attributed to the genomic variations in the EPS biosynthesis. The monosaccharide composition of EPS by *Bacillus cereus* AR156 was found to consist of mannose (70.97%), galactose (17.59%) and glucose (11.45%) [55], whereas EPS produced by a strain of seaweed–associated *Bacillus licheniformis* confirmed the presence of glucose (54.38%), mannose (25.24%) and galactose (11.32%), but also arabinose (9.06%) [56]. On the other hand, Farag et al. [57] reported the presence of a heteropolysaccharide with biomedical applications produced by *Bacillus mycoides* composed mainly of galactose, mannose and glucose, but also glucuronic acid.

#### 3.2.3. ATR–FTIR Analysis for EPS_t_

ATR–FTIR spectra (Figure 4) obtained from EPS_t_ showed peaks between 4000 cm^−1^ and 400 cm^−1^, characteristic of carbohydrates and the absence of proteins. The broad peak of O–H groups from 2900–3600 cm^−1^ and peaks at 1664 cm^−1^ and 1442 cm^−1^ could be attributed to the C=O and C–O stretch of the COO groups, respectively. Similar results were reported by the EPS produced by *Bacillus cereus* KMS3–1, composed of mannose, rhamnose, glucose and xylose, which showed characteristics peaks at 1635 and 1404 cm^−1^ attributed to the C=O and C–O stretch of the COO groups [58]. Similar results were reported from an EPS of *Lactobacillus plantarum* JLAU103, composed of different monosaccharides, including arabinose, rhamnose, fucose, xylose, mannose, fructose, galactose and glucose, with an absence of uronic acids and proteins. This EPS showed an absorption peak at 1423.06 cm^−1^ characteristic of carboxyl groups or carboxylates, further indicating that the EPS of *Lactobacillus plantarum* JLAU103 was an acidic polysaccharide [59]. The presence of these carboxyl groups may serve as binding sites for divalent cations [60]. IR bands 1052 cm^−1^ in the region of 950–1200 cm^−1^ correspond to CO and CC stretching vibrations in carbohydrates. The region from 1000–500 cm^−1^ is the fingerprint region and is unique to each molecule. The IR bands in the region of 800–950 cm^−1^ are very sensitive to the anomeric configuration of glucose. The band at 863 cm ^−1^ was found in both spectra, indicating an α- configuration of the glucose unit [61]. 

#### 3.2.4. Characterisation of the Thermal Properties of EPS_t_

The characterisation of the thermal properties of EPS_t_ was carried out by thermogravimetric analysis (TGA); the results are shown in Figure 5a. EPS_t_ experienced two states of weight loss as a function of temperature increase. The first stage with an initial weight loss was observed between 25 °C and 150 °C, with a weight loss of 33.28%, typical of the loss of moisture, indicating that the EPS could have a certain number of carboxyl groups, which would increase the affinity of the polysaccharide to its interaction with water. In addition, the appearance of these groups in the EPS could be an environmental adaptation to acidic environments [62]. The second stage presented a gradual weight loss of approximately 5.62% at 150–500°C, reaching stability from this point, which could be due to the EPS_t_ monosaccharide composition. Similar thermal properties were described for different *Bacillus* EPSs. In the case of the EPS produced by *Bacillus licheniformis* [56], composed of glucose, mannose, galactose and arabinose, two steps of degradation were also experienced. There was an initial weight loss (7%) of moisture from 30 to 120 °C followed by a second stage of degradation (59.6%), with a maximum loss at ≥330 °C. In addition, otherstudies with *Bacillus*-related species [63] demonstrated that EPS produced by *Lactobacillus paracasei* M7, composed of mannose, glucose and galactose, experienced an initial phase of weight loss (18%) at 95 °C due to moisture loss and a second sharp degradation phase at 210 °C until the polysaccharide was fully degraded at 480 °C. 

On the other hand, energy levels of the polysaccharide were scanned using a differential scanning calorimeter (DSC). The DSC thermogram (Figure 5b) showed an exothermic peak with a crystallisation temperature (Tc) of 70.15 °C. Similar results can be found with the EPS isolated from *Bacillus anthracis* PFAB2 [64], composed only of glucose, exhibiting an exothermic peak with a crystallisation temperature (Tc) of 90.67 °C. In this way, thermal properties of EPS_t_ indicated its potential stability for high-temperature industrial processes.

### 3.3. Biotechnological Applications

#### 3.3.1. Antioxidant Activity Tests 

The antioxidant activity (Figure 6) of EPS_t_ was measured in the concentration range of 0.1 to 10 mg/mL using the colourimetric assay described in Section 2.6. Ascorbic acid was used as the positive control. The DPPH free radical scavenging is shown in Figure 6a. The measured average scavenging activity of EPS_t_ was around 65% for all of the concentrations tested. In comparison, the scavenging activity of the EPS from *Bacillus amyloliquefaciens* GSBa−1 on DPPH radicals showed activities of 14.74 ± 1.02% at low concentrations [65]. Furthermore, the DPPH radical scavenging activity showed an increased tendency with a concentration (from 0.1 mg/mL to 2 mg/mL) of levan-type EPS produced by *Bacillus subtilis* AF17 in the range of 27.96 ± 1.2% to 58.54 ± 2.92% [66]. Free radical scavenging activity may be due to the presence of hydroxyl groups and other functional groups in the EPS. More stable forms can be obtained by the cession of an electron by EPS_t_ or by the reaction with free radicals to complete a radical chain reaction. The DPPH inhibitory potential of the EPS_t_ suggested that this may have sufficient proton donors that can react with free radicals to convert them to stable molecules [67].

The results for hydroxyl radical scavenging activity are shown in Figure 6b. The scavenging activity grew gradually with the increase in EPS_t_ concentration. At a concentration of 2.5 mg/mL, EPS_t_ reached 100% hydroxyl radical scavenging activity. The EPS_t_ exopolysaccharide showed a strong hydroxyl radical scavenging activity that might be due to the ability of EPS’ hydroxyl groups to donate active hydrogen [68]. In addition to this, the EPS_t_ had a higher scavenging activity than those from the EPSs produced by *Lactobacillus delbureckii* subsp. *bulgaricus* and *Bacillus aerophilus* rk1 EPS, reaching 67.5% and 78.6% at 4 mg/mL and 10 mg/mL, respectively [69,70]. This could be due to the bond dissociation energy of EPS_t_ being relatively weak; thereupon, it was easy to provide more energy or electron atoms to bind to the hydroxyl radical.

Figure 6c shows the scavenging activities in vitro on the superoxide anion of EPS_t_. The highest activity (39.4%) was found at the highest EPS_t_ concentration assayed. In contrast, EPS isolated by *Bacillus thuringiensis* RSK CAS4 presenting anticancer activities showed significant superoxide radical scavenging activity at 1.0 mg/mL of EPS (75.12%), being much more efficient than EPS_t_ [71]. Other studies indicated that EPSs produced by *Bacillus tequilensis* FR9 isolated from chicken exhibited a maximum of 57.22 ± 7.6% antioxidant activity at a 4 mg/mL concentration [72]. The antioxidant activity of EPS_t_ might be dependent on the physical and chemical structure of the biomolecules [73].

The metal ion chelating activity in Figure 6d,e was essayed with different EPS_t_ concentrations (0.1–10 mg/L) and pHs (5.6 and 2.5). The results significantly showed (*p* < 0.05) that EPS_t_ presented a total capacity of iron chelation over 5 mg/mL at pH 5.6 (Figure 6d) and 10 mg/mL at pH 2.5 (Figure 6e). These results are higher than those shown with the EPS isolated from *Lactobacillus helveticus* MB2–1 [74], which exhibited a chelating capacity on Fe^2+^ at 4.0 mg/mL of up to 59.11%. In contrast, the results reported from crude EPS, EPS−1 and EPS−2 produced by *Paenibacillus polymyxa* EJS−3 at concentrations of 1 mg/mL were 92.4%, 81.1% and 86.5%, respectively [75]. However, metal chelating capacities for these EPSs were not tested at low pHs. The metal chelating activity of EPS_t_ could be due to the antioxidation capacity that the EPS_t_ presented as a consequence of the polysaccharide’s chelating properties for metals. The EPS_t_ could produce a reduction in the transition metal concentration [76]. In addition to this, the EPS_t_ could reduce the redox potential as it stabilised the oxidated form of the metallic ion [77]. In addition, microorganisms can release products rich with COO− (carboxyl group) and OH− (hydroxyl ion) that act as chelators of cations such as Fe [78].

#### 3.3.2. Biocompatibility Studies and Antioxidant Ability on Cellular Level

Biocompatibility results of EPS_t_ showed that there was no significant damaging effect on HeLa cells after treatment with all of the concentrations of EPS_t_ tested. The results were not significantly different (*p* < 0.05), demonstrating that EPS_t_ is non–cytotoxic against HeLa cells. In comparison, results reported from the EPS produced by *Lactobacillus plantarum* WLPL04 [79] significantly decreased the cellular viability of human intestinal epithelia cells Caco–2 at concentrations of 200 and 400 μg/mL. Moreover, the EPS isolated from *Enterococcus faecalis* [80] was cytotoxic against HeLa cells in a dose-dependent manner, which killed almost 68.60% of HeLa cells at a 500 µg/mL concentration. The antioxidant ability of EPS_t_ at a cellular level is represented in Figure 7. Figure 7a shows the cell viability of HeLa cells exposed to different concentrations of H_2_O_2_. A reduction in cell viability under 60% was observed due to the accumulation of reactive oxygen species (ROS), which resulted in oxidative stress in the cells. To determine the antioxidant effect of EPS_t_ over HeLa cells (Figure 7b), the cell viability of HeLa cells was evaluated at different EPS_t_ concentrations. The results demonstrated that EPS_t_ concentrations between 25–200 µg/mL had a statistically significant improvement on cellular viability. In comparison, similar results [81] reported that the pretreatment of EPS (0.25, 0.50, 1 mg/mL) by *Weissella cibaria* significantly protected the HEK293 cells from H_2_O_2_–induced oxidative stress compared to untreated HEK293 cells. This is an indication that EPS_t_ was efficient in eliminating the excess of free radicals due to the high levels of ROS caused by H_2_O_2_ [82] whilst also having a protecting effect [83]. In the cell system, antioxidant enzymes such as superoxide dismutase (SOD) and catalase (CAT) have the largest effect in eliminating the excessive free radicals. This suggests that EPS_t_ could stimulate the antioxidant system, such as promoting the ability of the antioxidant enzymes [40].

#### 3.3.3. Emulsifying Properties 

The emulsifying properties of EPS_t_ are shown in Figure 8. In Figure 8a,b, the emulsifying properties of the exopolysaccharide EPS_t_ are shown when used with natural oils (olive, sunflower, coconut and sesame) at concentrations of 1 mg/mL (Figure 8a) and 3 mg/mL (Figure 8b) at 24 (E24) and 168 (E168) hours. After 24 h, at the lowest concentration of EPS_t_ (1 mg/mL), EPS_t_ showed a significantly (*p* < 0.05) high efficiency when emulsifying olive and sesame oils (reaching 90% and 83.3%, respectively); however, this was not as effective with sunflower and coconut oil (5% and 22.2%, respectively). At 168 h, a decrease in activity was detected for all treated natural oils. For olive oil, it decreased by 23%, whereas, for sesame oil, the effectiveness decreased by 46%. The emulsifying activity at the highest concentration of EPS_t_, 3 mg/mL, followed the same tendency as the 1 mg/mL concentration: a higher activity with sesame and olive oil but not as effective with sunflower and coconut. At 24 h, emulsification was overall lower with olive and sesame oils (55% and 60%, respectively) but higher with sunflower and coconut in comparison to 1 mg/mL (15% and 2%, respectively). After 168 h, emulsification for sesame oil was more stable than at 1 mg/mL as it remained over 50%, whereas, for other oils, the activity was lower overall (notably, sunflower oil showed no emulsification activity). The emulsifying capacity was significantly higher (p < 0.05) at 24 h than at 168 h, indicating that its capacity was moderate.

The exopolysaccharide EPS_t_ was compared with commercial emulsifiers (Triton X−100, Tween 20 and SDS). This is shown in Figure 8c (1 mg/mL),d (3 mg/mL). At both concentrations, the efficiency with olive and sesame oils reached greater emulsifying properties in comparison to Triton X−100 and SDS controls at the same concentrations. Tween 20 was more effective for olive oil than the EPS_t_ (5% more effective for 1 mg/mL and 35.9% more effective for 3 mg/mL) but, for sesame oil, the EPS_t_ was 18% and 45.7% more effective at 1 mg/mL and 3 mg/mL. Commercial emulsifiers were generally more effective in emulsifying sunflower and coconut oil.

In general, EPS_t_ showed a greater emulsifying effect at low concentrations (1 mg/mL). EPS_t_ had similar results to *Bacillus anthracis* EPS, which showed maximum emulsification activity (E24) for olive oil (95.3%), followed by sesame oil (90%) [65]. EPS_t_ had superior results to *Bacillus amyloliquefaciens* EPS, which had emulsifying properties for olive oil (E24) of 58.58% [84]. 

It is considered in the literature that a good emulsifier can maintain at least 50% of the original volume of the emulsion after 24 h [85]. Although the emulsification stabilities of both exopolysaccharide concentrations (1 mg/mL and 3 mg/mL) showed a gradually decreasing trend with time, the emulsification stabilities of both concentrations reached 50% for both olive and sesame oil, showing a relatively good emulsification stability. 

EPS_t_ showed significantly better emulsifier properties when compared to commercial emulsifiers as observed in Figure 6c,d. In this case, the EPS’s molecular structure, specific concentration and functional groups could have influenced the emulsifying properties. These factors have been found to have a considerable effect on the emulsifying properties of polysaccharides [86]. The EPS_t_ polysaccharide is a good candidate for the substitution of the most commercially used and contaminating emulsifiers, such as the more efficient Tween 20. Tween 20 belongs to a polysorbate group that can cause inflammatory effects associated with processed foods [87]. On the other hand, EPS_t_ was found to not be cytotoxic and, due to its emulsifying properties, it could well be used as an emulsifying agent in pharmaceutical and other industries [88,89,90]. 

## 4. Conclusions

An extreme-tolerant *Bacillus xiamenensis* RT6 strain was isolated from Río Tinto (Huelva), Spain. A comparison between different growth mediums (MGM with glucose and an enriched medium) determined that the highest production of EPS took place with the glucose medium as the only carbon source of energy, reaching an EPS production of up to 700 mg/mL, and also demonstrating that EPS production was not positively correlated with a higher cellular growth. The molecular weight of EPS_t_ was estimated to be 2.71 × 10^4^ Da, showing a slightly lower molecular weight in comparison with most heteropolysaccharides. Polymer characterisation revealed that it was a heteropolysaccharide with good thermal stability, composed mainly of glucose (60%), mannose (20%) and galactose (20%). Structural analysis suggested the presence of O–H, C=O and C–O groups of EPS_t_. In the case of application, in vitro antioxidant studies showed that EPS_t_ has a strong potential as a natural polymer to act as an antioxidant with strong scavenging activities of radicals (DPPH and hydroxyl radicals), also biocompatible at cellular level, enhancing protection to ROS damaged cells. EPS_t_ revealed good metal chelating activities at high concentrations, also applicable at low pHs (2.5), showing significant differences against lower concentrations. Finally, emulsifying properties indicated that EPS_t_ has an excellent emulsifying capacity against some natural oils (olive and sesame). Furthermore, these results suggest the exopolysaccharide’s suitability for industrial applications such as bioremediation processes of natural oils, food emulsification processes and cosmetic or pharmaceutical applications as an antioxidant. 

## Figures and Tables

**Figure 1 polymers-14-03918-f001:**
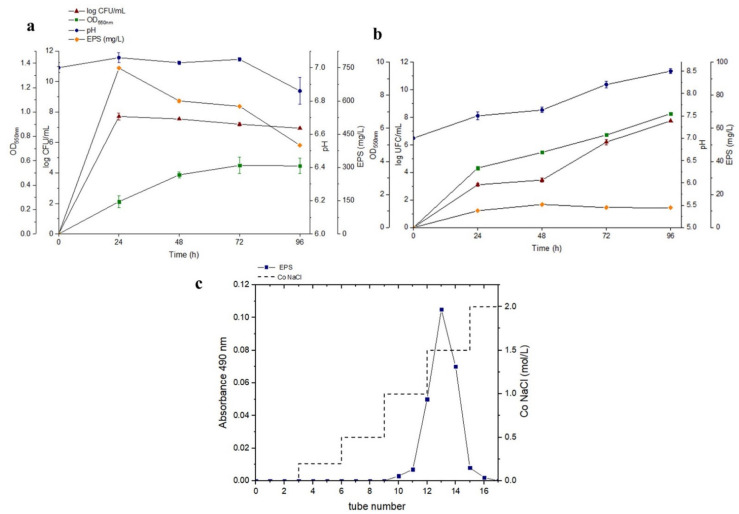
(**a**) Analysis of bacterial growth, CFU/mL, pH and EPS production in MGM with glucose. (**b**) Analysis of bacterial growth, CFU/mL, pH and EPS production in an enriched medium. (**c**) Elution curve obtained from purification of EPS_t_ extracted from bacteria grown on MGM with glucose.

**Figure 2 polymers-14-03918-f002:**
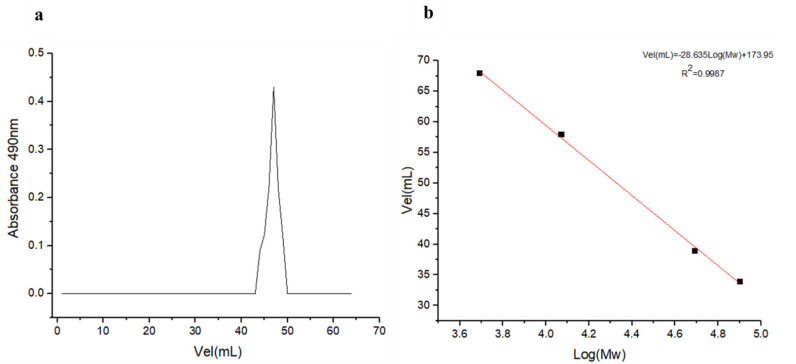
Molecular weight estimation for EPS_t._ (**a**) Elution curve of EPS_t_ by Sephadex G−100 gel filtration. (**b**) Standard curve of the relative molecular weight (Mw).

**Figure 3 polymers-14-03918-f003:**
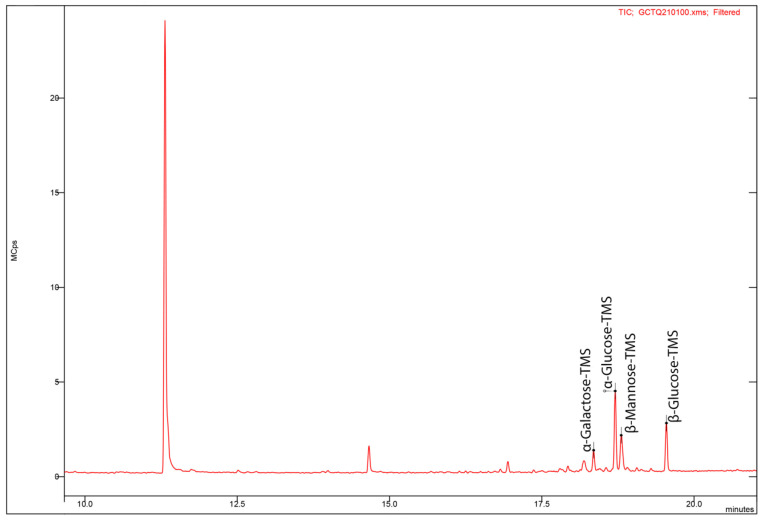
GC–MS gas chromatography analysis for characterisation of EPS_t_.

**Figure 4 polymers-14-03918-f004:**
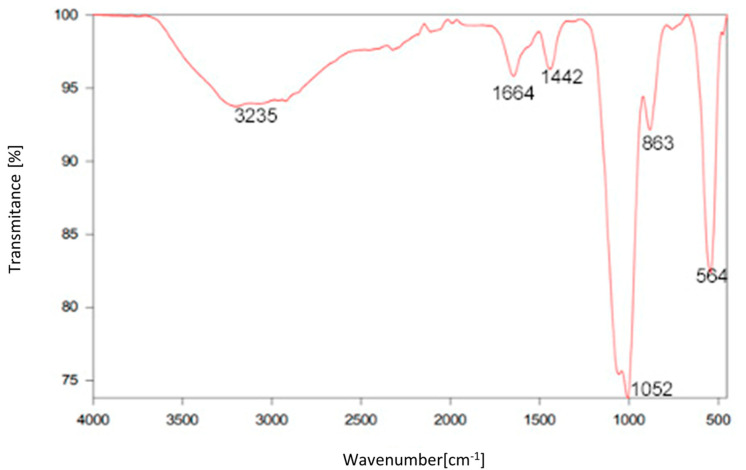
ATR-FTIR spectra for compositional characterisation of EPS_t_.

**Figure 5 polymers-14-03918-f005:**
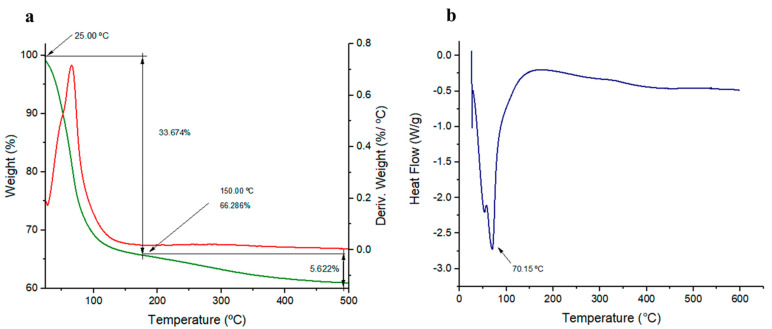
(**a**) Thermogravimetric (TGA) analysis of EPS_t_. (**b**) Differential scanning calorimetry (DSC) analysis for EPS_t_.

**Figure 6 polymers-14-03918-f006:**
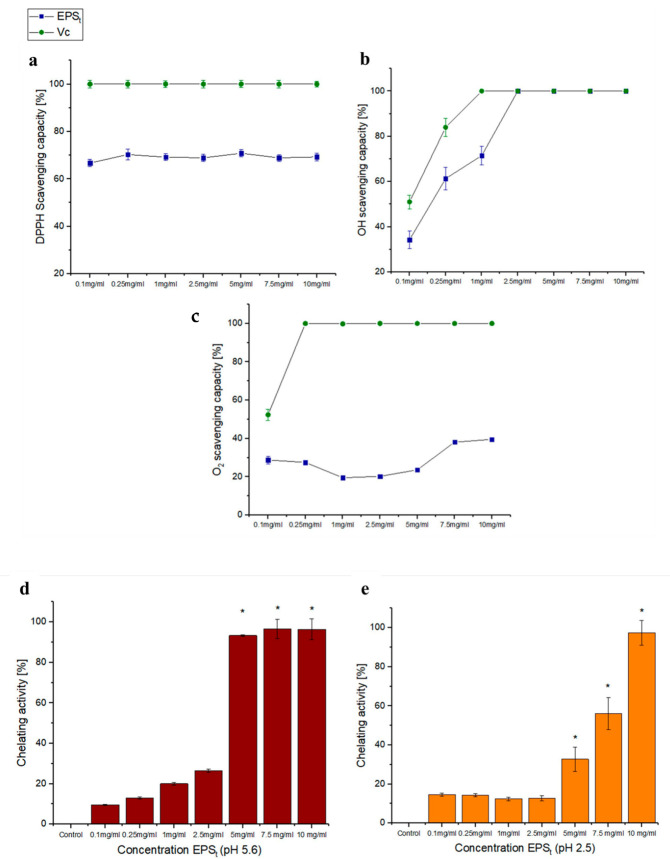
Antioxidant tests in vitro with different concentrations of EPS_t_. (**a**) DPPH free radical scavenging activity of EPS_t_ compared to that of ascorbic acid (Vc). (**b**) Hydroxyl radical scavenging activity of EPS_t_ compared to that of ascorbic acid (Vc). (**c**) Superoxide anion scavenging activity of EPS_t_ compared to that of ascorbic acid (Vc). (**d**) EPS_t_ chelating activity at different concentrations measured at pH 5.6. (**e**) EPS_t_ chelating activity at different concentrations measured at pH 2.5. (n = 3, significant differences * *p* < 0.05).

**Figure 7 polymers-14-03918-f007:**
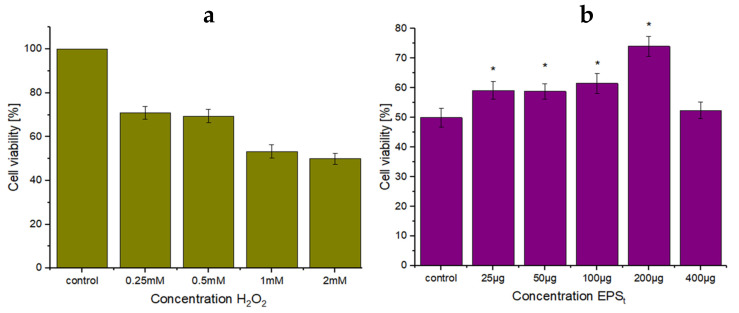
(**a**) HeLa cells viability (%) against oxidative stress by different H_2_O_2_ concentrations. (**b**) Exhibition of EPS_t_ protection on HeLa cells viability (%). (n = 3, significant differences * *p* < 0.05.)

**Figure 8 polymers-14-03918-f008:**
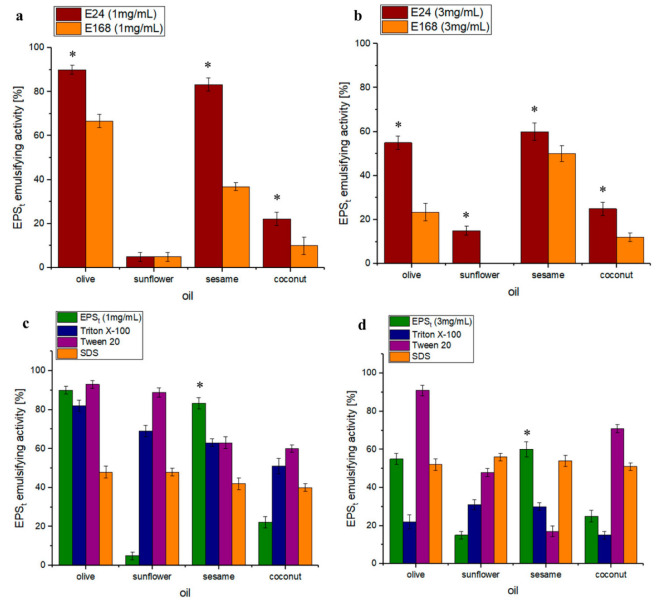
(**a**) Emulsifying properties with 1 mg/mL of EPS_t_. The percentage of emulsion of EPS_t_ with the different oils used at 24 h (E24) and 168 h (E168) of study. (**b**) Emulsifying properties with 3 mg/mL of EPS_t_. The percentage of emulsion of EPS_t_ with the different oils used at 24 h and 168 h of study. (**c**) Comparison of the emulsifying properties with 1 mg/mL of EPS_t_ against commercial emulsifiers (Triton X−100, Tween 20 and SDS) across different natural oils. (**d**) Comparison of the emulsifying properties with 3 mg/mL of EPS_t_ against commercial emulsifiers (Triton X−100, Tween 20 and SDS) across different natural oils. (n = 3, significant differences * *p* < 0.05).

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
