# Peer review of "Potential Applications of an Exopolysaccharide Produced by Bacillus xiamenensis RT6 Isolated from an Acidic Environment"

_polymers, 2022, doi:10.3390/polym14183918_

Round 1

Reviewer 1 Report

This study reports the average molecular weight and evaluated the antioxidant capabilities of an exopolysaccharide (EPS) derived fromBacillus xiamenensis RT6 strain. More interestingly, EPSt was able to emulsify different natural polysaccharide oils and was a good candidate for the substitution of the most polluting emulsifiers.

These findings are interesting.The manuscript can be accepted after minor revision.

Author Response

Thank you for your review, we're glad that you've found the study interesting.

Reviewer 2 Report

No novelty and similar studies published.

Author Response

Thank you for your review. We have now amended the introduction which should make the novel aspects of the study more apparent:
"This study aims to describe the isolation and identification by morphological, biochemical and molecular biology of the Bacillus xiamenensis RT6 strain isolated from an extremely acidic environment, Río Tinto (Huelva, Spain). In addition to EPS production optimisation, molecular weight estimation and characterization by GC-MS, HPLC/MSMS, ATR-FTIR, TGA and DSC; its biotechnological applications were evaluated. This species has been described in agricultural applications as a potential PGPR (Plant-Growth Promoting Rhizobacteria) antagonist [28] and biocontrol agent [29]. However, the strain is novel and its biotechnological applications have not been studied. The hypothesis of this work is that Bacillus xiamenensis RT6 may have a wide range of potential biotechnological applications, such as pharmaceutical and industrial use. Therefore, the present study was carried out to test different capacities including anti-oxidation, metal chelation, cell viability and emulsification properties."

Reviewer 3 Report

Dear authors,

This paper deals with the isolation and characterization of Exopolysaccharide produced by a bacterial strain isolated under extreme acid conditions. There are several matters that need to be attended to before being considered for publication.

The abstract and introduction are informative and well written. However, it is not clear why the collection site is extremely acid?.... there is no information about that in the manuscript.... consider show evidence or removing the phrase in title and manuscript.

Methodology section.

For all equipment, it must be mentioned the Brand name, company name, city, and country.

Line 127...Before the last sentence, maybe you can add a phrase saying the EPS will be isolated and quantified for each treatment...

Line 143...Molecular weight determination must be placed in the compositional analysis below Line 148... Also, which Dextran standards were used for Mw?, who was the supplier?... add to the material section.

Line 170... 4cm-1 of resolution...check grammar

Line 238...which type of cell was used...It is not well clear how the cellular antioxidant activity was determined...check this complete section. Is there any mathematical equation?. Check other papers because there are other names for this type of assay... this is not truly a cellular antioxidant test.

Line 252...Fe chelating ability is a known antioxidant method, then it is suggested to move this method to the corresponding antioxidant activity.

Result and Discussion section

Line 323 to 326... proper comparison of EPS Mw with other EPS with a similar composition, or from similar bacterial strains, must be done...that's why it is suggested this section can be placed after EPS composition.

As a general rule, figures must be placed near the first time they were mentioned to make it easiest to follow the discussion.

I suggest discussing each characterization technique in a separate manner for the sake of clarity.

Line 336-343 ...when comparing with other EPS compositions add the sugar ratio found for each EPS.

Improved discussion of ATR must be done...there are other relevant bands that may be associated with -NH groups?... The C=O groups ... are associated with which compound or functional groups...where do they come from? proteins? no mention of them inside results was done, but in the methodology section there is a mention.

Figure 2. The GC-MS picture (a) is repeated...check this. Separate each analysis in different graphics.

It is suggested the TGA picture 2c, must be started from 25 to 500°C (x-axes) and up to 101% (Y axes). Then the DTG must be arranged.

In the case of TGA, the DTG graph must be added in addition to the DSC one. Thus consider to built two graphs. An improved discussion (initial degradation temperature or temperature degradation range must be added. Comparison of thermal stability with other EPS with similar compositions is required. I don't think this EPS melts, maybe you can state the EPS decomposition at this point is exo or endothermic...The melting process is not associated with weight loss, it is just a physical method. The thermal decomposition process associated with this EPS must be better explained. 

Figure 3...no letters are associated with graphs...

Figure 4 can be merged with figure 6...all antioxidant activity. Discussion of the antioxidant capacity must be improved.

Discussion of figure 5 must be improved dramatically. I am not sure if you can state the in vitro antioxidant assay to this one...because there is a specific bioassay for this purpose. Check this.

if you performed statistical analysis then you must discuss it in the manuscript.

It is suggested to change emulsifying activity by emulsifying properties. It s clear the emulsifying capacity was higher at 24h than 168h , indicating its capacity is moderate...

conclusions must be improved

Author Response

Thank you for your review, please find our responses below:

Reviewer #3: This paper deals with the isolation and characterization of Exopolysaccharide produced by a bacterial strain isolated under extreme acid conditions. There are several matters that need to be attended to before being considered for publication.

My comments:  

  1. The abstract and introduction are informative and well written. However, it is not clear why the collection site is extremely acid?... there is no information about that in the manuscript.... consider show evidence or removing the phrase in title and manuscript.

For further clarification, we added on page 2 line 57 the information regarding why the collection site is extremely acid in the manuscript´s introduction.

“On the other hand, Rio Tinto is considered as an extreme environment exhibiting a constant very acidic pH (2.5) buffered by ferric iron and with high concentrations of heavy metals. These extreme conditions are originated by the metabolic activity of chemolithotrophic prokaryotes that are found in high numbers in its waters [27]”

Added bibliography

 https://doi.org/10.1016/j.aquatox.2005.09.002.

  1. For all equipment, it must be mentioned the Brand name, company name, city, and country.

All equipment´s brand name, company, city and country are added to the manuscript:

Page 2 line 75. BBLTM Sensi-DiscTM (Sensi-Disc™, Fisher Scientific, MA USA)

Page 2 line 77. Hela cells, CLS (Cell Line Service, Germany).

Page 2 line 79. API gallery, BioMerieux, España S.A.

Page 3 line 85. Trypticase Soya Agar (TSA) plates (Sigma-Aldrich Ireland Limited, Arklow, Ireland)

Page 3 line 89. Microscope B100iMSa, Spain

Page 3line 91. API gallery (BioMérieux, Spain).

Page 3 line 94. Commercial discs (BBL™ Sensi-Disc™, Fisher Scientific, MA, USA)

Page 3 line 102. FastDNA kit for soil (BIO 101, Vista, Calif) 

Page 3 line 103. JetQuick® Genomic DNA Purification Kit (Genomed)

Page 3 line 108. Cycler 2720 Thermocycler (Applied Biosystems, Foster City, CA)

Page 3 line 112. Greensafe (NZYtech, Portugal).

Page 3 line 114. BLAST (National Centre of Biotechnology, Bethesda, MD, USA)

Page 4 line 129. Spectophotometer Biowave II+ (Biochrom Cambridge, UK).

Page 4 line 133. Thermo Orion pH Meter Model 2Star (Thermo Scientific, Asheville, NC)

Page 4 line 138. Rotary shaker incubator (model Orbitek LJEIL; Scigenics Biotech, Bangalore, India)

Page 4 line 149. DuPont Sorvall RC-5 centrifugue (Wilmington, DE)

Page 4 line 153. Flexy-Dry MPTM freeze dryer (FtS Systems Inc., Stone Ridge NY, USA)

Page 4 line 157. DEAE-52 anion exchange column (Aldrich Chemical Company, Inc., USA)

Page 5 line 166. Standard reference dextrans of 5, 12, 50 and 80 KDa (Sigma, St. Louis, MO, USA)

Page 5 line 167. Sephadex G-100 column (Aldrich Chemical Company, Inc., USA)

Page 5 line 172. Trifluoroacetic Acid (TFA) Aldrich® (Steinheim, Germany)

Page 5 line 176. EVOQ GC-TQ Bruker (Bruker, America) gas chromatography system.

Page 5 line 184. HPLC/MSMS (Agilent Technologies 1100 series ‐ 6410B (TQ), Waldbronn, Germany)

Page 5 line 185. ACE Excel 3 C18-Amide column (Advanced Chromatography Technology LTD, Aberdeen, Scotland)

Page 5 line 191. BX-FTIR spectrometer (Perkin Elmer, USA)

Page 5 line 192. MIRacleTM -ATR (Pike Technologies, Cottonwood, USA)

Page 5 line 197. TGA Q500 (TA Instruments, New Castle, DE)

Page 6 line 202. DSC Q100 (TA Instruments, New Castle, DE

Page 6 line 211. FLUOstar Omega BMG LABTECH (Aylesbury, UK)

Page 6 line 217. DPPH (100 μM DPPH-ethanolic solution) (Sigma Chemical, St Louis, MO)

Page 6 line 219. Ascorbic acid (Vc) (Sigma Chemical, St Louis, MO)

Page 6 line 234. Salicylic acid (9 mM ethanol-salicylic acid solution) (Sigma Chemical, St Louis, MO)

Page 6 line 236. H2O2 (8.8 mM) (Sigma Chemical, St Louis, MO)

Page 7 line 253. Pyrogallol (3 mM) (Sigma Chemical, St Louis, MO)

Page 7 line 269. Ferrozine solution (5 mM) (Sigma Chemical, St Louis, MO)

Page 7 line 273. Ethylenediaminetetraacetic acid (EDTA) (Sigma Chemical, St Louis, MO)

Page 8 line 288. Dulbecco’s Modified Eagle’s Medium (DMEM) (Sigma Chemical, St Louis, MO)

Page 8 line 289.  Fetal Bovine Serum (FBS) (Sigma Chemical, St Louis, MO)

Page 8 line 289. L-glutamine (Sigma Chemical, St Louis, MO)

Page 8 line 290. Penicillin (100 IU/mL) (Sigma Chemical, St Louis, MO)

Page 8 line 291. Streptomycin (100 μg/mL) (Sigma Chemical, St Louis, MO)

Page 8 line 296. MTT reagent (GE Healthcare, Uppsala, Sweden)

Page 8 line 300. Microplate reader (LT-4000, Labtech International Ltd, UK)

Page 8 line 311. MTT method (GE Healthcare, Uppsala, Sweden)

Page 8 line 327. MTT method (GE Healthcare, Uppsala, Sweden)

Page 9 line 342. Polysorbate 20 (Tween 20) (Sigma Chemical, St Louis, MO)

Page 9 line 343. Sodium Dodecyl Sulphate (SDS) (Sigma Chemical, St Louis, MO)

Page 9 line 344. Triton X-100 (Sigma Chemical, St Louis, MO)

  • Line 127...Before the last sentence, maybe you can add a phrase saying the EPS will be isolated and quantified for each treatment...

Now on page 4 line 143. We’ve added the phrase for more clarity “the EPS will be isolated and quantified for each treatment”

  1. Line 143...Molecular weight determination must be placed in the compositional analysis below Line 148.

This is now on page 4 line 164. We have rearranged the structure of this part and now the determination of the molecular weight of EPSt section has been moved to belongs to section “2.5.1 Compositional analysis” under “2.5 Compositional analysis and characterization of the EPS”

  1. Which Dextran standards were used for Mw?, who was the supplier?... add to the material section.

This is now on page 5 line 166. All Dextran standards selected for molecular weight determination were acquired from Sigma, St. Louis, MO, USA. For the experiment, Dextran standards of different molecular weight (5, 12, 50 and 80 KDa) were used.

  1. Line 170... 4cm-1 of resolution...check grammar.

This is now on page 5 line 193. The grammar of the sentence has bee corrected. “…interferograms were obtained from 32 scans at 4 cm–1 of resolution from 400 to 4000 cm–1

  • Line 238...which type of cell was used...

This is now on page 8 line 320. Hela cells acquired from CLS Cell Line Service (Germany) were used for this experiment. For a better understanding, we added the cell line name at the beginning of the section: “Hela cells density were readjusted to 5 x 104 CFU/well and the suspension was seeded into a 96-well plate for 24 h”, “The determination of the protection capacity of EPSt on Hela cells against oxidative stress was assayed following the methods described by Zhou et al. [40]. Hela cells were seeded into 96-well plate for 24 h at a concentration of 5 x 104 CFU/well.”

  • It is not well clear how the cellular antioxidant activity was determined...check this complete section. Is there any mathematical equation? Check other papers because there are other names for this type of assay... this is not truly a cellular antioxidant test.

This is now on page 8 line 306. For better clarification, we introduced further the explanation and we included a reference of Zhou et al [40]  https://doi.org/10.1016/j.arabjc.2020.11.007.

Also on page 9 line 304. The MTT method determines the antioxidant activity quantitatively. For instance, with this method, we subjected the cells into oxidative stress by adding H2O2. It is reported that H2O2 could induce intracellular oxidative stress, not only by direct oxidant injury, but also contributing to cytotoxic effects on cells by the reaction with peroxidases to form highly reactive molecules or activating signaling pathways to stimulate ROS production on cells. All resulting in the quick accumulation of ROS, production of free radicals such as O- and consequently, oxidant injury. In our study, oxidative stress could be confirmed by the decrease on cell viability [%] measured with the MTT method previously described.

In this way, we planned a second assay by adding EPSt to pretreated Hela cells, then, the cells were exposed to H2O2. In this way we could evaluate EPSt antioxidant capacity against the oxidative stress produced by H2O2 by again, the MTT method. For further clarification, on page 8 line 306 and page 8 line 321, we have added a reference to support our method used: Zhou et al., [40] https://doi.org/10.1016/j.arabjc.2020.11.007.

For more clarification, we added on page 8 line 314 and page 8 line 331 a mathematical equation.

“Hela cells viability was calculated according to Formula 5:

Cell viability [%] = (A1/A2) x 100                              (5)

Where,

A1 = absorbance at OD 590 nm of the cells with MTT solution treated previously with H2O2 (and EPSt for Formula 6)

A2 = absorbance at OD 590 nm of the cells with MTT solution without any treatment”

Added bibliography

  1. Line 252...Fe chelating ability is a known antioxidant method, then it is suggested to move this method to the corresponding antioxidant activity.

Now on page 7 line 264. We moved this part to section “2.6.4 Metal ion chelating activity” under “2.6 Antioxidant activity tests”

Result and Discussion section

  1. Line 323 to 326... proper comparison of EPS Mw with other EPS with a similar composition, or from similar bacterial strains, must be done...that's why it is suggested this section can be placed after EPS composition.

This is now on page 11 line 408-414. As requested, we included more discussion to the molecular weight result from EPSt by comparing it with other Bacillus EPSs. Also we compared our result with an average range of Mw previously described for most of heteropolysaccharides.

“In this case, EPSt presented a lower molar mass in comparison with most of the reported heteropolysaccharides [51] https://doi.org/10.1016/S0958-6946(01)00160-1. Studies showed that the average molecular weight of an EPS produced by freshwater dynamic sediment-attached Bacillus megaterium RB-05 [52] https://doi.org/10.1111/j.1365-2672.2011.05162.x, composed of glucose, galactose, mannose, arabinose, fucose and N-acetyl glucosamine, was found to be 1.7 x 105 Da. Moreover, EPS isolated from Bacillus coagulans RK-02 [53] https://doi.org/10.1021/np1008448, composed of galactose, mannose, fucose, and glucose, had an average molecular weight of 3 × 104 Da”

Added bibliography

  1. That's why it is suggested this section can be placed after EPS composition.

Now on page 11 line 404. We moved this section to “3.2.1 Molecular weight determination for EPSt” under “3.2 Compositional analysis and characterisation of EPSt exopolymer”

  • As a general rule, figures must be placed near the first time they were mentioned to make it easiest to follow the discussion.

For a better understanding, we separated the compositional analysis and characterisation EPSt exopolymer section figures.

Now Figure 2 represents the molecular weight estimation for EPSt.

Now Figure 3 represents the GC-MS gas chromatography analysis for characterisation of EPSt..

Now Figure 4 represents the ATR-FTIR spectra for compositional characterisation of EPSt.

Now Figure 5 represents the Thermogravimetric (TGA) and Differential Scanning Calorimetry (DSC) analysis for EPSt.

  • I suggest discussing each characterization technique in a separate manner for the sake of clarity.

We have separated each characterisation technique into subsections for a better understanding:

Now on page 11 line 404. “3.2.1 Molecular weight determination for EPSt

Now on page 12 line 419. “3.2.2 Gas chromatography GC-MS analysis for EPSt

Now on page 13 line 457.  “3.3.3 ATR-FTIR analysis for EPSt

Now on page 14 line 484.  “3.2.4 Characterisation of the thermal properties of EPSt

  • Line 336-343 ...when comparing with other EPS compositions add the sugar ratio found for each EPS.

Now on page 12 Line 424-434. We added a further discussion for the EPSs compared in this section adding each monomer total content (%):

“Other Bacillus strains such as the EPS produced by Bacillus velezensis SN-1 with antioxidant and antitumor properties reported the presence of mannose and glucose in an approximate of 54.60 % and 45.30 %, respectively, but an absence of galactose [54]. This variability in content can be attributed to the genomic variations in the EPS biosynthesis. The monosaccharide composition of EPS by Bacillus cereus AR156 found to consist of mannose (70.97 %), galactose (17.59 %) and glucose (11.45 %) [55] https://doi.org/10.3389/fmicb.2016.00664.. Whereas, EPS produced by a strain of seaweed associated Bacillus licheniformis confirmed the presence of glucose (54.38 %), mannose (25.24 %), galactose (11.32 %), but also arabinose (9.06%) [56] https://doi.org/10.1016/j.carbpol.2010.12.061. On the other hand, Farag et al. [57] reported the presence of an heteropolysaccharide with biomedical applications produced by Bacillus mycoides composed mainly of galactose, mannose, glucose, but also glucuronic acid.”

Added bibliography

  1. Improved discussion of ATR must be done...there are other relevant bands that may be associated with -NH groups?...The C=O groups ... are associated with which compound or functional groups...where do they come from? proteins? no mention of them inside results was done, but in the methodology section there is a mention.

Now on page 12 line 423- 424. We included the following statement in the results to indicate the absence of proteins.

 “In addition, results obtained in the determination of amino acids and uronic acids by HPLC, confirmed the absence of both found in the sample”

We have discharged the presence of proteins, so for instance, the association with -NH groups.

Now on page 13 line 458. We have included for further clarity:

 “ATR-FTIR spectra (Figure 4) obtained from EPSt showed peaks between 4000 cm-1 and 400 cm-1, characteristic of carbohydrates and the absence of proteins”

Now on page 13 line 460. We added further discussion:

“The broad peak of O-H groups from 2900-3600 cm-1, peaks at 1664 cm-1 and 1442 cm-1 could be attributed to the C=O and C-O stretch of the COO groups, respectively [58, 59]”

supported by the following reference: https://doi.org/10.1016/j.carbpol.2019.115369.

  • Figure 2. The GC-MS picture (a) is repeated...check this.

Now Figure 3. GC-MS figure has been corrected.

  • Separate each analysis in different graphics.

Compositional and structural analysis are now separated into different figures.

Now Figure 2 represents the molecular weight estimation for EPSt..

Now Figure 3 represents the GC-MS gas chromatography analysis for characterisation of EPSt.

Now Figure 4 represents the ATR-FTIR spectra for compositional characterisation of EPSt..

Now Figure 5 represents the Thermogravimetric (TGA) and Differential Scanning Calorimetry (DSC) analysis for EPSt.

  • It is suggested the TGA picture 2c, must be started from 25 to 500°C (x-axes) and up to 101% (Y axes). Then the DTG must be arranged.

Now Figure 5. Done.

  • In the case of TGA, the DTG graph must be added in addition to the DSC one.

Now Figure 5a. DTG is already added to the TGA graph.

  1. Thus consider to built two graphs. An improved discussion (initial degradation temperature or temperature degradation range must be added. Comparison of thermal stability with other EPS with similar compositions is required. I don't think this EPS melts, maybe you can state the EPS decomposition at this point is exo or endothermic...The melting process is not associated with weight loss, it is just a physical method. The thermal decomposition process associated with this EPS must be better explained.

Now Figure 5. We have separated TGA and DSC graphs for a better comprehension. Now they are separated as: Figure 5a. Thermogravimetric (TGA) analysis of EPSt and Figure 5b. Differential Scanning Calorimetry (DSC) analysis for EPSt.

Now page 14 line 485. We have introduced more relevant data and an improve discussion for this section:

“The characterisation of the thermal properties of EPSt was carried out by thermogravimetric analysis (TGA), the results are shown in Figure 5a. EPSt experienced two states of weight loss as a function of temperature increase. The first stage with an initial weight loss was observed between 25 °C and 150 °C, with a weight loss of 33.28 %, typical of the loss of moisture, indicating the EPS could have a certain number of carboxyl groups, which would increase the affinity of the polysaccharide for its interaction with water. In addition, the appearance of these groups, in the EPS, could be an environmental adaptation to acidic environments [62]. The second stage presented a gradually weight loss of approximately 5.62 % at 150- 500°C, reaching stability from this point, which could be due to the EPSt monosaccharide composition. Similar thermal properties were described for different Bacillus EPSs. In the case of the EPS produced by Bacillus licheniformis [63] https://doi.org/10.1016/j.carbpol.2010.12.061, composed of glucose, mannose, galactose and arabinose, experienced also two steps of degradation. There was an initial weight loss (7 %) of moisture from 30 to 120 °C followed by a second stage of degradation (59.6 %) with maximum loss at ≥ 330 °C. Also, other studies with Bacillus related species [64] https://doi.org/10.1016/j.bcdf.2019.100191 demonstrated that EPS produced by Lactobacillus paracasei M7, composed of mannose, glucose and galactose, experienced an initial phase of weight loss (18 %) at 95 °C due to moisture loss and a second sharped degradation phase at 210 °C till the polysaccharide was fully degraded at 480 °C.

On the other hand, energy levels of the polysaccharide were scanned using a differential scanning calorimeter (DSC). DSC thermogram (Figure 5b) showed an exothermic peak with crystallization temperature (Tc) of 70.15 °C. Similar results can be found with the EPS isolated from Bacillus anthracis PFAB2 [65], https://doi.org/10.1007/s12088-017-0699-4, composed only of glucose, exhibited an exothermic peak with crystallization temperature (Tc) of 90.67 °C. In this way, thermal properties of EPSt indicated its potential stability for high temperature industrial processes”

Added bibliography

  • Figure 3...no letters are associated with graphs...

Now Figure 6. Letters have been included.

  • Figure 4 can be merged with figure 6...all antioxidant activity.

Now Figure 6. We merged the antioxidant assays (DPPH, Hydroxyl and Superoxide anion scavenging activities) with the metal chelating properties assay.

  • Discussion of the antioxidant capacity must be improved.

We added more discussion by comparing our results with other studies of Bacillus strains EPSs.

This is on page 15 line 519-521.

Furthermore, the DPPH radical scavenging activity showed an increased tendency with concentration (from 0.1 mg/mL to 2 mg/mL) of levan type EPS produced by Bacillus subtilis AF17 in the range of 27.96 ± 1.2 % to 58.54 ± 2.92 % [67] https://doi.org/10.1016/j.ijbiomac.2019.12.108.”

And on page 16 line 539-543.

“In contrast, EPS isolated by Bacillus thuringiensis RSK CAS4 presenting anticancer activities showed significant superoxide radical scavenging activity at 1.0 mg/mL of EPS (75.12 %), being much more efficient than EPSt [72]. Other studies indicated that EPS produced by Bacillus tequilensis FR9 isolated from chicken exhibited a maximum of 57.22 ± 7.6% antioxidant activity at 4 mg/mL concentration [73] https://doi.org/10.1016/j.ijbiomac.2016.11.122.”

Also on page 16 line 550-554. “These results are higher to those showed with the EPS isolated from Lactobacillus helveticus MB2-1 [75] https://doi.org/10.1016/j.lwt.2014.06.063,  which exhibited chelating capacity of on Fe2+ at 4.0 mg/mL up to 59.11 %. In contrasts, results reported from crude EPS, EPS-1 and EPS-2 produced by Paenibacillus polymyxa EJS-3 at concentrations of 1 mg/mL were 92.4 %, 81.1 % and 86.5 %, respectively [76] https://doi.org/10.1016/j.fct.2011.11.016. However, metal chelating capacities for these EPSs were not tested at low pHs.”

Now on page 16 line 558-560. “Also, microorganism can release products rich with COO− (carboxyl group) and OH− (hydroxyl ion) which act as chelators of cations such as Fe [79]   https://doi.org/10.3390/app11094160 ”

Added bibliography

  • Discussion of figure 5 must be improved dramatically. I am not sure if you can state the in vitro antioxidant assay to this one...because there is a specific bioassay for this purpose. Check this.

Now Figure 7. We added further discussion for a better understanding:

Now on page 18 line 571- 578.

“Biocompatibility results of EPSt showed that there was no significant damaging effect on Hela cells after treatment with all the concentrations of EPSt tested. In this way, results were not significantly different (p <0.05), demonstrating that EPSt is non-cytotoxic against Hela cells. In comparison, results reported from the EPS produced by Lactobacillus plantarum WLPL04 [80] https://doi.org/10.3168/jds.2018-15831 significantly decreased the cellular viability of human intestinal epithelia cells Caco-2 at concentrations of 200 and 400 μg/mL. Moreover, the EPS isolated from Enterococcus faecalis [81] https://doi.org/10.1007/s00284-020-02130-z was cytotoxic against Hela cells in a dose-dependent manner, which killed almost 68.60 % of Hela cells at 500 µg/mL concentration”

Now on page 18 line 584-588.

“The results demonstrated that EPSt concentrations between 25-200 µg/mL had a statistically significant improvement on cellular viability. In comparison, similar results [82] https://doi.org/10.1016/j.lwt.2021.112727 reported that the pretreatment of EPS (0.25, 0.50, 1 mg/mL) by Weissella cibaria significantly protected the HEK293 cells from H2O2 induced oxidative stress compared to untreated HEK293 cells.”

Now on page 18 line 588- 591.

 “In the cell system, antioxidant enzymes such as superoxide dismutase (SOD) and catalase (CAT) have the largest effect in eliminating the excessive free radicals. This suggests that EPSt could stimulate the antioxidant system such as promoting the ability of the antioxidant enzymes [40] https://doi.org/10.1016/j.arabjc.2020.11.007.”

Added bibliography

  • If you performed statistical analysis, then you must discuss it in the manuscript.

For further clarity, we added more discussion to the results with statistical analysis:

Now on page 15 line 567.

 “(n=3, significant differences *p < 0.05)”

Now on page 15 line 571.

“Biocompatibility results of EPSt showed that there was no significant damaging effect on Hela cells after treatment with all the concentrations of EPSt tested.”

Now on page 15 line 572.

“The results were not significantly different (p <0.05), demonstrating that EPSt is non-cytotoxic against Hela cells.”

Now on page 15 line 584.

“The results demonstrated that EPSt concentrations between 25-200 µg/mL had a statistically significant improvement on cellular viability.”

Now on page 16 line 598.

“(n=3, significant differences *p < 0.05)”

Now on page 16 line 605.

EPSt showed significantly (p < 0.05) high efficiency when emulsifying olive and sesame oils (reaching 90 % and 83.3 % respectively)”

Now on page 16 line 615.

“The emulsifying capacity was significantly higher p < 0.05 at 24 h than 168 h, indicating that its capacity is moderate.”

Now on page 21 line 653.

(n=3, significant differences * p < 0.05)

No on page 22 line 669.

“EPSt revealed good metal chelating activities at high concentrations, also applicable at low pHs (2.5), showing significant differences against lower concentrations.”

  • It is suggested to change emulsifying activity by emulsifying properties.

We changed emulsifying activity into emulsifying properties now on page 19 lines 600, 601, 619, 629, 634, 636, 638, 639 and 643.

  • It’s clear the emulsifying capacity was higher at 24h than 168h, indicating its capacity is moderate...

Now on page 19 line 616. We have included the sentence.

  • Conclusions must be improved

Now on page 22 line 654-674. We improved the conclusion including more data obtained from the study:

“An extreme-tolerant Bacillus xiamenensis RT6 strain was isolated from Río Tinto (Huelva), Spain. A comparison between different growth mediums (MGM with glucose and an enriched medium) determined that the highest production of EPS took place with the glucose medium as the only carbon source of energy, reaching an EPS production up to 700 mg/mL, also demonstrating that EPS production was not positively correlated with a higher cellular growth. The molecular weight of EPSt was estimated to be 2.71 x 104 Da, showing a slightly lower molecular weight in comparison with most heteropolysaccharides. Polymer characterisation revealed that it was an heteropolysaccharide with good thermal stability, composed mainly of glucose (60 %), mannose (20 %) and galactose (20 %). Structural analysis suggested the presence of O-H, C=O and C-O groups of EPSt. In the case of application, in vitro antioxidant studies showed that EPSt has a strong potential as a natural polymer to act as an antioxidant with strong scavenging activities of radicals (DPPH and hydroxyl radicals), also biocompatible at cellular level, enhancing protection to ROS damaged cells. EPSt revealed good metal chelating activities at high concentrations, also applicable at low pHs (2.5), showing significant differences against lower concentrations. Finally, emulsifying properties indicated that EPSt has an excellent emulsifying capacity against some natural oils (olive and sesame). Furthermore, these results suggest the exopolysaccharide’s suitability for industrial applications such as bioremediation processes of natural oils, food emulsification processes and cosmetic or pharmaceutical applications as an antioxidant.”.

Reviewer 4 Report

The manuscript by Lin et al. reports interesting biotechnological aspects of exopolysaccharides produced by Bacillus species. 

I have strong recommendation for this article to be accepted after minor srevision. 

The data is original, technical aspects are taken into consideration and the way of presentation is fine. 

I would like the authors to add a hypothesis, overtaken by a strong research gap statement in the introduction section. 

Then, kindly add an overview of exoploysaccharides producing microbes and their application in agriculkture and environmental sustainability to support your objectives. 

Clearly state the study objectioves.

The authors might take help from the following publication;

Rhizosphere bacteria in plant growth promotion, biocontrol, and bioremediation of contaminated sites: a comprehensive review of effects and mechanisms

A review on practical application and potentials of phytohormone-producing plant growth-promoting rhizobacteria for inducing heavy metal tolerance in crops

Efficacy of indole acetic acid and exopolysaccharides-producing Bacillus safensis strain FN13 for inducing Cd-stress tolerance and plant growth promotion in Brassica juncea (L.)

The methods needs clarification on the aspects of antibiotic sensity, lines 86-99, This must be supported by suitable references. 

I suggest authors to give only the salient research findings and the rest should go in supplementary files. 

Author Response

Thank you for your review, please find our responses below:

Reviewer #4: The manuscript by Lin et al. reports interesting biotechnological aspects of exopolysaccharides produced by Bacillus species. I have strong recommendation for this article to be accepted after minor revision. The data is original, technical aspects are taken into consideration and the way of presentation is fine.

My comments:

  1. I would like the authors to add a hypothesis, overtaken by a strong research gap statement in the introduction section. Then, kindly add an overview of exopolysaccharides producing microbes and their application in agriculture and environmental sustainability to support your objectives. Clearly state the study objectives.

We have further clarified the objectives and hypothesis of the study to the introduction section.

Now on page 2 line 61-72. We have further exposed the objectives for a better understanding: “This study aims to describe the isolation and identification by morphological, biochemical and molecular biology of the Bacillus xiamenensis RT6 strain isolated from an extremely acidic environment, Río Tinto (Huelva, Spain). In addition to EPS production optimisation, molecular weight estimation and characterization by GC-MS, HPLC/MSMS, ATR-FTIR, TGA and DSC; its biotechnological applications were evaluated. This species has been described in agricultural applications as a potential PGPR (Plant-Growth Promoting Rhizobacteria) antagonist [28] and biocontrol agent [29]. However, the strain is novel and its biotechnological applications have not been studied. The hypothesis of this work is that Bacillus xiamenensis RT6 may have a wide range of potential biotechnological applications, such as pharmaceutical and industrial use. Therefore, the present study was carried out to test different capacities including anti-oxidation, metal chelation, cell viability and emulsification properties.”

2. The authors might take help from the following publication

Following your recommendations, we took help from these publications and we included them as the following:

Now on page 2 line 41,

 “Moreover, toxic levels of heavy metals may interact with important biomolecules in the cell, including DNA and protein, leading to excessive production of ROS [10]”

Now on page 2 line 69,

 “This species has been described in agricultural applications as a potential PGPR (Plant-Growth Promoting Rhizobacteria) antagonist …[28]

Now on page 16 lines 558-560,

 “Also, microorganisms can release products rich with COO− (carboxyl group) and OH− (hydroxyl ion) which act as chelators of cations such as Fe [79].

  • The methods need clarification on the aspects of antibiotic sensibility, lines 86-99, This must be supported by suitable references.

Now on page 3 line 92. We added a reference to support the method used in order to clarify the antibiotic sensibility of the strain.  https://doi.org/10.1128/jcm.32.10.2505-2509.1994

3. I suggest authors to give only the salient research findings and the rest should go in supplementary files.

We considered that it is not necessary to include our results at supplementary files.

Round 2

Reviewer 2 Report

The authors cannot responded to all comments

Author Response

Thank you

Reviewer 3 Report

Dear authors,

Most of the questions or comments were answered in a satisfactory manner. However, there is one comment and one question remaining.

This is now on page 5 line 193....you must change the word ... interferogram by .... spectra.

In FTIR spectra, you stated there are some bands corresponding to C=O and COO groups... I agree with you. But, if there are no proteins and no uronic acid in the sample, where do they come from?. which compound or functional group in your sample is responsible for this signal? comment on this.

Author Response

Most of the questions or comments were answered in a satisfactory manner. However, there is one comment and one question remaining.

1. This is now on page 5 line 193....you must change the word ... interferogram by .... spectra.

Now on page 5 line 193. We changed the word interferogram by spectra.

“…and spectra were obtained from 32 scans at 4 cm–1 of resolution from 400 to 4000 cm–1.”

2. In FTIR spectra, you stated there are some bands corresponding to C=O and COO groups... I agree with you. But, if there are no proteins and no uronic acid in the sample, where do they come from?. which compound or functional group in your sample is responsible for this signal? comment on this.

Now on page 13 line 461-468. For further clarity, we have included an extended discussion of our results according to the results and discussion reported by Krishnamurthy et al. (2020):

 https://doi.org/10.1016/j.carbpol.2019.115369 and Min et al. (2019): https://doi.org/10.1016/j.jbiosc.2018.12.004.

“Similar results were reported by the EPS produced by Bacillus cereus KMS3-1, composed of mannose, rhamnose, glucose, and xylose, showed characteristics peaks at 1635 and 1404 cm−1 attributed to the C=O and C-O stretch of the COO groups [58]. Similar results were reported from an EPS of Lactobacillus plantarum JLAU103, composed of different monosaccharides including arabinose, rhamnose, fucose, xylose, mannose, fructose, galactose, and glucose, with absence of uronic acids and proteins. This EPS showed an absorption peak at 1423.06 cm−1 characteristic of carboxyl groups or carboxylates, further indicating that EPS of Lactobacillus plantarum JLAU103 was an acidic polysaccharide [59].”

In addition to this, on page 16 line 565, it is reported that these COO− (carboxyl group) and OH− (hydroxyl ion) can act as chelators of cations such as Fe [79].